# Immune Responses to SARS-CoV-2 Infection and Vaccination in Dialysis Patients and Kidney Transplant Recipients

**DOI:** 10.3390/microorganisms10010004

**Published:** 2021-12-21

**Authors:** Patrick Affeldt, Felix Carlo Koehler, Karl August Brensing, Vivien Adam, Julia Burian, Linus Butt, Martin Gies, Franziska Grundmann, Steffen Hinrichs, Wibke Johannis, Nils Kalisch, Matthias Meyer-Delpho, Simon Oehm, Eva Platen, Claudia Schöler, Eva Heger, Gertrud Steger, Dirk Stippel, Aileen Ziegelhöfer, Thomas Benzing, Florian Klein, Christine Kurschat, Roman-Ulrich Müller, Veronica Di Cristanziano

**Affiliations:** 1Department II of Internal Medicine and Center for Molecular Medicine Cologne, Faculty of Medicine and University Hospital Cologne, University of Cologne, 50937 Cologne, Germany; patrick.affeldt@uk-koeln.de (P.A.); felix.koehler@uk-koeln.de (F.C.K.); vivien.adam@uk-koeln.de (V.A.); linus.butt@uk-koeln.de (L.B.); franziska.grundmann@uk-koeln.de (F.G.); steffen.hinrichs1@uk-koeln.de (S.H.); nils.kalisch@uk-koeln.de (N.K.); simon.oehm@uk-koeln.de (S.O.); thomas.benzing@uk-koeln.de (T.B.); christine.kurschat@uk-koeln.de (C.K.); 2CECAD, Faculty of Medicine and University Hospital of Cologne, University of Cologne, 50931 Cologne, Germany; 3Nierenzentrum Bonn, 53175 Bonn, Germany; dr.brensing@nephrologie-bonn.de; 4Nieren- und Dialysezentrum Rheinbach, 53359 Rheinbach, Germany; aerzte@ndz-rhb.de; 5KfH-Nierenzentrum Köln-Longerich, 50737 Cologne, Germany; martin.gies@kfh-dialyse.de (M.G.); claudia.schoeler@kfh-dialyse.de (C.S.); 6Institute for Clinical Chemistry, Faculty of Medicine and University Hospital of Cologne, University of Cologne, 50937 Cologne, Germany; wibke.johannis@uk-koeln.de; 7Dialysezentrum Siegburg, 53721 Siegburg, Germany; m.meyerdelpho@gmail.com; 8Nierenzentrum Eifel, 53894 Mechernich, Germany; eva.platen@web.de; 9Institute of Virology, Faculty of Medicine and University Hospital of Cologne, University of Cologne, 50935 Cologne, Germany; eva.heger@uk-koeln.de (E.H.); gertrud.steger@uk-koeln.de (G.S.); aileen.ziegelhoefer@uk-koeln.de (A.Z.); florian.klein@uk-koeln.de (F.K.); 10Department of General, Visceral, Cancer and Transplant Surgery, Faculty of Medicine, University Hospital Cologne, University of Cologne, 50937 Cologne, Germany; dirk.stippel@uk-koeln.de

**Keywords:** COVID-19, immunosuppression, protection, titer, antibodies, kidney disease

## Abstract

Dialysis patients and kidney transplant (KTX) recipients suffer from an impaired immune system and show a decreased response to the severe acute respiratory syndrome coronavirus type 2 (SARS-CoV-2) vaccination. We performed a retrospective analysis of 1505 serological SARS-CoV-2 measurements obtained from 887 dialysis patients and 86 KTX recipients. The results were separated by patient subgroups (dialysis/KTX) as well as SARS-CoV-2 status. The latter criterion included SARS-CoV-2-naïve patients with or without COVID-19 vaccination and convalescent patients receiving a booster shot. Serologies of 27 vaccinated healthy individuals served as the reference group. Vaccine-induced cellular immune response was quantified by an interferon-γ release assay in 32 KTX recipients. We determined seroconversion rates of 92.6%, 93.4%, and 71.4% in dialysis patients vaccinated with either BNT162b2, mRNA-1273, or AZD1222, respectively. Vaccination-induced anti-SARS-CoV-2 antibody titers were lower in dialysis patients compared to healthy individuals, and vaccination with mRNA-1273 induced higher titers than BNT162b2. The initial seroconversion rate was 39.5% in KTX recipients vaccinated with BNT162b2. A linear regression model identified medication with mycophenolate-mofetil/mycophenolic acid as an independent risk factor for missing seroconversion. Within a cohort of 32 KTX recipients, cellular and humoral immune reactivity to SARS-CoV-2 was detectable in three patients only. Conclusively, vaccine-induced seroconversion rates were similar in dialysis patients compared to healthy individuals but were strongly impaired in KTX recipients. Anti-SARS-CoV-2 IgG titers elicited by double active immunization were significantly lower in both cohorts compared to healthy individuals, and immune responses to vaccination vanished quickly.

## 1. Introduction

Hemodialysis (HD) patients and kidney transplant (KTX) recipients are at risk for severe coronavirus disease-19 (COVID-19) [1,2]. Surprisingly, although limited in number, observational studies have revealed no difference in immune response to the severe acute respiratory syndrome coronavirus type 2 (SARS-CoV-2) following natural infection in chronic immunosuppressed patients and immunocompetent individuals [3]. Apparently, age and frailty as well as the number and severity of pre-existing comorbidities—more than pharmacological immunosuppression—are prognostic factors for a severe COVID-19 disease course in both immunocompromised and immunocompetent patients [4,5]. Notwithstanding, European public health data indicate chronic kidney disease (CKD) and dialysis as critical prognostic comorbidities associated with increased mortality upon SARS-CoV-2 infection [6]. In addition, impaired seroconversion and longevity of SARS-CoV-2 immune responses after infection further threatens HD and KTX patients [4,7,8].

Active immunization against SARS-CoV-2, in particular with messenger ribonucleic acid (mRNA) vaccines, is considered the most effective preventive strategy to protect vulnerable individuals from infection and severe COVID-19 courses [9]. However, both the intensity and duration of the immune response after vaccination in specific patient groups has not been fully characterized. Compared to the general population, HD patients are considered to present diminished SARS-CoV-2 antibody titers and have a delayed immune response [10,11,12]. After the second injection, seroconversion rate may be as high as 90% and a cellular immune response can be detected in approximately 80% of vaccinated individuals [13,14,15]. The increasing available data from different studies on the efficacy of SARS-CoV-2 vaccination in solid organ transplant (SOT) recipients revealed a strongly impaired humoral and cellular immune response compared to the general population, with a reported seroconversion of below 45% and additionally an impaired cellular response [16,17,18,19].

Thus far, the definition of an antibody titer conferring protection is still being intensively investigated and the determination of antibody levels is not routinely performed [20]. We systematically analyzed the seroconversion rate and antibody titer in 681 HD patients and 86 KTX recipients to characterize the humoral response to both PCR-confirmed SARS-CoV-2 infection and vaccination (BNT162b2 (BioNTech/Pfizer, Mainz, Germany/New York, NY, USA), mRNA-1273 (Moderna, CA, USA) and AZD1222 (AstraZeneca/University of Oxford, Cambridge/Oxford, UK)) in a retrospective manner. Serologies of 27 mRNA-vaccinated healthy individuals served as a reference group. Furthermore, the cellular immune response was prospectively quantified in 32 KTX recipients to fully characterize mRNA vaccine-induced protection.

## 2. Materials and Methods

### 2.1. Study Design, Patients, and Ethical Statement

To characterize the humoral response to both infection and vaccination, we retrospectively analyzed SARS-CoV-2 serologies from dialysis and KTX patients performed at the University Hospital Cologne, Germany, starting at the beginning of the COVID-19 pandemic in April 2020 until June 2021. This cohort contained 206 SARS-CoV-2-naïve dialysis patients in which anti-SARS-CoV-2 antibodies were screened every three months for one year from April 2020 to April 2021. Regarding vaccination, antibody measurements obtained between March 2021 and June 2021 from 627 SARS-CoV-2-naïve dialysis patients, of whom 475 received BNT162b2 (BNT162b2 (BioNTech/Pfizer, Mainz, Germany/New York, NY, USA), 138 mRNA-1273 (Moderna, CA, USA), and 14 AZD1222 (AstraZeneca/University of Oxford, Cambridge/Oxford, UK), respectively, were available. In addition, serologies of 54 dialysis patients who had recovered from SARS-CoV-2 were obtained, of whom 38 received a booster shot (all BNT162b2). Regarding SARS-CoV-2-naïve KTX recipients, sera was obtained from 86 patients to monitor BNT162b2 vaccination. Serologies of 27 healthy individuals vaccinated with mRNA-1273 served as a control group. All vaccines were administered in clinical routine at the discretion of the treating physician and as advised by the manufacturers. All of the measurements were performed in clinical routine and analyzed retrospectively after approval from the local institutional review board (Ethics Committee of the Medical Faculty of the University of Cologne, Germany, EK 21-1398). Additionally, humoral and cellular immune responses were examined prospectively in 32 SARS-CoV-2-naïve KTX recipients between June and September 2021. This study was also approved by the Ethics Committee of the Medical Faculty of the University of Cologne, Germany (EK 21-1112) and was conducted in line with the Declaration of Helsinki.

### 2.2. Serological Assays for the Detection of Anti-SARS-CoV-2 Antibodies

Anti-SARS-CoV-2 IgG targeting the spike (S) protein was detected by the semiquantitative Euroimmun anti-SARS-CoV-2 IgG ELISA using the Euroimmun Analyzer I (Euroimmun Diagnostik, Lübeck, Germany), and by the quantitative IDK^®^ anti-SARS-CoV-2 IgG ELISA (Immundiagnostik AG, Bensheim, Germany) using the DYNEX DSX^®^ (Dynex Technologies, Chantilly, VA, USA), both using the recombinant S1 antigen of the spike protein, or the chemiluminescent microparticle immunoassay by Abbott for the quantitative detection of anti-spike RBD (receptor binding domain) IgG (SARS-CoV-2 IgG II Quant) using the automated system Alinity i (Abbott, Abbott Park, IL, USA). The results of the quantitative IgG assays targeting S1 or RBD were expressed in BAU/mL (binding antibody unit) using the conversion factor generated from a World Health Organization (WHO) internal standard and provided by each manufacturer. Additionally, 29 patients from the hemodialysis group were parallel examined using these two quantitative SARS-CoV-2 IgG assays to determine if they delivered comparable IgG titers. Nucleocapsid (NC) protein-targeting antibodies were detected by the pan-immunoglobulin immunoassay Elecsys^®^ anti-SARS-CoV-2 electrochemiluminescence immunoassay (ECLIA) using the Cobas 8000 (Roche Diagnostics, Mannheim, Germany). All assays were interpreted according to the manufacturers’ recommendations.

### 2.3. Determination of T-Cell Response to SARS-CoV-2

The CD4+ and CD8+ T-cell-mediated response to SARS-CoV-2 was measured by the commercial whole blood interferon-γ (IFN-γ) release assay (IGRA) QuantiFERON^®^ SARS-CoV-2 RUO (QIAGEN GmbH, Hilden, Germany). T-cells were stimulated with epitopes of S1 and S2 subunits of SARS-CoV-2 spike protein (antigen 1 and 2). IFN-γ concentration was measured by a chemiluminescence immunoassay (CLIA), as previously described [21]. An IFN-γ level ≥ 0.15 IU/mL was scored as reactive [22].

### 2.4. Statistics

Statistical analyses were performed using GraphPad Prism software version 5 (GraphPad Prism Software Inc. San Diego, CA, USA) and SPSS version 27.0 (SPSS Inc., Chicago, IL, USA). Non-normal distributions of antibody titers were expressed as median ± interquartile range and compared using the Kruskal–Wallis/Mann–Whitney *U* tests and Spearman Correlation. Regarding patient characteristics, categorical variables were summarized by absolute numbers and frequencies, and numerical variables by median and interquartile range, respectively. Categorical variables were analyzed using a chi-square test, and numerical variables using a Mann–Whitney *U* test in a descriptive manner. Correlations—adjusted for used assay, age, and time between event and blood sample—were calculated using a linear regression model with Spearman analysis. *p* < 0.05 was considered statistically significant. To examine the comparability of the two immunoassays, a Passing–Bablok regression was carried out using the software Analyze-it v5.50 (Analyse-it Software, Ltd. Leeds, UK).

## 3. Results

### 3.1. Humoral Immune Response in SARS-CoV-2-naïve Dialysis Patients after Infection

In order to monitor SARS-CoV-2 infections, a cohort of 206 SARS-CoV-2-naïve dialysis patients was screened for anti-SARS-CoV-2 NC antibodies one year after the beginning of the pandemic, using a pan-Ig assay. In the surveillance period between April 2020 and April 2021, ten clinically apparent SARS-CoV-2 infections, including three fatal outcomes, were detected (Figure 1A). Approximately 40 days after initial SARS-CoV-2 diagnosis by real-time PCR, antibodies were identified in 71.4% (5/7) of the dialysis patients that recovered from COVID-19. Of interest, one hemodialysis patient tested as reactive for anti-SARS-CoV-2 NC antibodies without clinically apparent COVID-19.

### 3.2. Humoral Response in SARS-CoV-2-naïve Dialysis Patients after Vaccination

A total of 627 SARS-CoV-2-naïve dialysis patients were immunized by double vaccination, with 475 receiving BNT162b2, 138 mRNA-1273, and 14 AZD1222, respectively (Figure 1B). The time between vaccination and blood sampling for serology was approximately 30 days and did not differ between the different groups. Seroconversion was detected in 92.6% (440/475) of the vaccinated dialysis patients receiving BNT162b2 and in 93.4% (129/138) receiving mRNA-1273 (Table 1).

There was no significant difference between the two mRNA-based vaccines regarding seroconversion. Genetic nephropathies, other than autosomal dominant polycystic kidney disease (ADPKD), were linked with lack of seroconversion (*p* = 0.003) (Table 2).

In contrast, neither age, sex, other pre-existing kidney diseases, nor mode of or time on dialysis influenced vaccine-induced seroconversion in our dialysis cohort. Compared to mRNA-1273-vaccinated healthy individuals, levels of anti-SARS-CoV-2 antibodies upon vaccination with either BNT162b2, mRNA-1273, or AZD1222 were significantly lower in dialysis patients (Figure 2A). Furthermore, there were significantly higher levels of mRNA-1273-induced anti-SARS-CoV-2 IgG titer compared to BNT162b2 (*p* = 0.001). In addition, a linear regression model revealed a highly significant correlation between higher IgG titers and use of the mRNA-1273 vaccine, even after adjusting for the type of immunoassay used and the time between sampling and vaccination (*p* = 0.001; Appendix A). However, the mRNA-1273-vaccinated dialysis cohort was also younger (*p* < 0.001) and more patients suffered from dialysis-dependent diabetic nephropathy (*p* = 0.002) (Appendix A). In contrast, the group of BNT162b2-vaccinated patients contained more patients on continuous ambulatory peritoneal dialysis (CAPD). Only 2.2% of dialysis patients in our cohort were immunized with vector-based AZD1222, and their seroconversion rate was 71.4% (10/14)—significantly lower compared to the mRNA-based vaccines (Table 1). Although small in number, AZD-1222-induced anti-SARS-CoV-2 IgG titers were not significantly different to mRNA-induced levels in dialysis patients (Figure 2A).

### 3.3. Booster Shot-Induced Humoral Response in Dialysis Patients after Recovery from COVID-19

Anti-SARS-CoV-2 serology was established in a cohort of 681 dialysis patients, with 54 patients having recovered from COVID-19. Additional booster shots (all BNT162b2) were administered in 70.4% (38/54) of convalescent dialysis patients (Figure 1C). Compared to healthy individuals vaccinated with mRNA-1273, convalescent dialysis patients with and without an additional booster shot revealed reduced anti-SARS-CoV-2 IgG values with no difference between the two groups (Figure 2B). It is of note that, compared to the mRNA vaccine in immune-naïve dialysis patients, booster-shot vaccination in recovered dialysis patients elicited higher anti-SARS-CoV-2 IgG levels (*p = 0.07*), even after adjustment for the assay used and the timespan between blood sampling and vaccination (*p* < 0.01) (Figure 2C, Appendix A).

### 3.4. Comparisons between Standardized Anti-RBD and Anti-S1 IgG Titers

To determine if SARS-CoV-2 IgG titers differed between immunoassays targeting RBD or S1 regions, median IgG titers were compared in 29 dialysis patients 30 days after their second vaccination with BNT162b2. For the RBD antigen-based assay, the median IgG titer was 390 BAU/mL (IQR 793), and for the S1 antigen it was 493.48 BAU/mL (IQR 875.74), with a high correlation between both IgG titers (Spearman correlation coefficient 0.99, *p* < 0.0001) (Appendix A). In addition, mean IgG titers did not differ significantly in these two groups (*p* = 0.658, Appendix A). A Passing–Bablok Regression was also performed to further investigate the comparability of the two quantitative serological assays and revealed sufficient comparability (intercept: 2.6 (95%-confidence interval (CI): −10.7 to 14.7); slope: 1.16 (95% CI: 1.0 to 1.3) (Appendix A).

### 3.5. Humoral Response in SARS-CoV-2-Naïve Kidney Transplant Recipients Quantified by Anti-SARS-CoV-2 Serology after Vaccination

Humoral response was characterized by anti-SARS-CoV-2 serology in 86 SARS-CoV-2-naïve KTX recipients immunized using BNT162b2 with an approximate gap of median 30 days between their second vaccination and sampling (Figure 1D). The serology-based seroconversion rate in KTX recipients was 39.5% (34/86). Compared to healthy individuals, vaccine-induced anti-SARS-CoV-2 titers were heavily impaired in KTX recipients (Figure 3A). 

Neither age, sex, nor transplantation mode (deceased or living) was linked to impaired seroconversion (Table 3). However, anti-SARS-CoV-2 IgG titers showed a tendency towards a positive correlation with the timespan between kidney transplantation and vaccination (Spearman correlation *p* = 0.05, Appendix A). The seroconversion rate and anti-SARS-CoV-2 IgG levels were significantly lower in KTX recipients on immunosuppressive regimens containing mycophenolate mofetil (MMF) or mycophenolic acid (MPA) (Table 3, Figure 3B). In addition, a linear regression model revealed a highly significant correlation between immunosuppressive regimens with MMF/MPA and lacking seroconversion (*p* < 0.001, adjusted for vaccine used, age, and time between sampling and vaccination; Appendix A).

### 3.6. Cellular Response in SARS-CoV-2-Naïve Kidney Transplant Recipients Quantified by Interferon-γ Release Assay after Vaccination

The vaccine-induced CD4+ and CD8+ T-cell-mediated cellular response in KTX recipients was quantified using the QuantiFERON SARS-CoV-2, with IFN-γ ≥ 0.15 IU/mL considered as positive. In the case of 32 KTX recipients, cellular and humoral reactivity were measured prospectively in parallel (Appendix A). After a median of 61 days after vaccination, anti-SARS-CoV-2 IgG was detected in two patients (patient 1: 13.7 BAU/mL measured 9 days after second BNT162b2 vaccination; patient 2: 99.4 BAU/mL measured 13 days after BNT162b2 vaccination). Only one patient showed a CD4+ T-cell response to the SARS-CoV-2 spike protein (antigen 1 = 0.21 IU/mL), whereas anti-SARS-CoV-2 IgG was not detectable after BNT162b2 vaccination in this patient. In addition, we did not find any vaccine-induced CD8+ T-cell responses in KTX recipients.

## 4. Discussion

Dialysis patients and KTX recipients have an impaired immune system and are at risk for SARS-CoV-2-related complications. In addition, infection risk in dialysis patients may be increased as social distancing is challenging in dialysis centers, and during travel to and from the centers [7,23,24,25]. Hemodialysis units themselves are therefore considered high-risk venues for SARS-CoV-2 spread to a vulnerable cohort [24]. With increasing knowledge during the pandemic and implemented precaution measures, transmission rates in dialysis centers have been decreasing [23,26]. In line with this notion, we reported ten SARS-CoV-2 infections—three with fatal outcome—in a German surveillance cohort of 206 dialysis patients in a one-year observational period from April 2020 to April 2021. Furthermore, we identified an asymptomatic seroconversion in only one dialysis patient, in contrast to observational studies from China and Spain that reported clinically inapparent seroconversion ranging from 24% to 47% [26,27].

Vaccination is a central tool to protect from SARS-CoV-2 infection. However, a diminished response to vaccination has been reported for dialysis patients, with neutralizing antibodies detected in approximately 80% of immunized dialysis patients [1,12,14]. We found a seroconversion rate after vaccination ranging from 71.4% in AZD1222 to 92.6% in BNT162b2 and 93.4% in mRNA-1273 vaccinated patients. Anti-SARS-CoV-2 serology was investigated in a cohort of 627 dialysis patients with additional 54 patients who had recovered from COVID-19. Of note, vaccine-induced seroconversion in our German dialysis cohort was similar to the seroconversion rate after vaccination in healthy individuals as identified in the pivotal studies [28,29,30]. In line with this notion, we observed higher seroconversion rates after mRNA-based immunization compared to vector-based vaccines. Data regarding viral vector-based vaccination in dialysis patients are scarce. Single-dose Ad26.COV2.S (Janssen-Cilag/Johnson & Johnson, Beerse, Belgium) vaccination resulted in a seroconversion rate of 37% among dialysis patients [31]. Seroconversion rates after two doses of AZD1222 in dialysis patients have not been reported yet; however, we established a seroconversion rate of 71.4%. Notwithstanding, the limited patient numbers receiving AZD1222 have to be acknowledged as a weakness in our study.

Interestingly, genetic nephropathies—other than ADPKD—were associated with diminished seroconversion in our cohort. In contrast, neither age, sex, other pre-existing kidney diseases, nor mode of and years on dialysis were linked to an impaired humoral response upon vaccination in our dialysis cohort. It is of note that, compared to healthy individuals, vaccine-induced anti-SARS-CoV-2 titers were strongly reduced in dialysis patients. Interestingly, mRNA-1273 induced higher anti-SARS-CoV-2 antibody titers compared to BNT162b2. However, the mRNA-1273-vaccinated dialysis cohort was younger than the BNT162b2 cohort and dialysis was more often due to diabetic nephropathy, which may additionally influence the observed antibody titers. Notwithstanding, our observational data point towards a stronger mRNA-1273-induced humoral immune response compared to BNT162b2 in SARS-CoV-2-naïve dialysis patients [16]. Interestingly, booster-shot vaccination in dialysis patients after recovery from infection resulted in higher anti-SARS-CoV-2 IgG titers compared to mRNA vaccine-induced titers in immune-naïve dialysis patients. Anti-S1 and anti-RBD IgG titers were highly correlated in a group of 29 hemodialysis patients one month after their second BNT162b2 vaccination, in line with previous findings [32]. In addition, a Passing–Bablok regression showed sufficient comparability between the two quantitative IgG assays used in the present analysis. However, the deviation between the two assays seems to increase with increasing levels of IgG titers. Especially for antibody titers > 10,000 BAU/mL, potentially higher S1-IgG than RBD-IgG titers could be measured. In our largest cohort of SARS-CoV-2-naïve dialysis patients, the assay selection had no influence on the IgG titers normalized in BAU/mL (linear regression model *p* = 0.281). However, in the significantly smaller group of SARS-CoV-2-naïve patients after KTX, a low significant correlation between the selected assays and the SARS-CoV-2 IgG titers normalized to BAU/mL was identified (*p* = 0.04). Nevertheless, all significant subgroup differences in SARS-CoV-2 IgG titers were still significant after adjusting for the selected assays. Due to the large number of different assays available, the question of the comparability of IgG titers normalized to BAU/mL has been repeatedly discussed [32]. Our investigations showed that the two assays used are in principle comparable and small measurement differences did not affect the conclusions, especially in the case of large group sizes.

While SARS-CoV-2-naïve dialysis patients showed a high vaccine-induced seroconversion rate, this rate was significantly impaired in KTX recipients. In our German cohort of 86 KTX recipients, BNT162b2 vaccine-induced seroconversion measured one month after vaccination was lacking in 60.5% (52/86) of patients. It is of note that recent results obtained in the RECOVAC Immune Response study point towards a higher seroconversion rate of 56.9% in KTX recipients vaccinated with two doses of mRNA-1273 compared to BNT162b2 [33,34]. Vaccine-induced anti-SARS-CoV-2 titers were markedly reduced in KTX recipients compared to healthy individuals. Our data are in line with recently reported data on the antibody response elicited by BNT162b2 in KTX recipients [17,18,19,35]. A linear regression model identified medication with mycophenolate mofetil (MMF)/mycophenolic acid (MPA) as a strong independent risk factor linked to missing seroconversion in our KTX cohort receiving BNT162b2. According to previous evidence, this finding may point towards pausing this medication in long-term stable KTX recipients around vaccination as a potential tool to increase seroconversion rates [16,34]. Overall, anti-SARS-CoV-2 IgG titers were significantly lower in KTX recipients on MMF/MPA in the case of BNT162b2 vaccination. There are hints that mRNA-1273 elicits stronger immune responses in KTX recipients [16,34,35,36]. In addition, we observed that vaccine-induced anti SARS-CoV-2 antibodies vanished rapidly in KTX recipients, as only 6.1% (2/32) of the prospectively monitored KTX recipients had detectable antibodies at a median of two months after BNT162b2 vaccination. Considering the impaired humoral response to vaccination in KTX recipients, cellular immunity could still provide significant protection, raising doubts that serological exams help when judging protection from SARS-CoV-2 infection in at-risk cohorts [36]. Cellular immunity has also been hypothesized to maintain longer-term protection, even when antibody levels have subsided [37]. Therefore, we determined the long-term cellular immune response using an IFN-γ release assay in addition to humoral reactivity in vaccinated KTX recipients. However, longer-term vaccine-induced anti-SARS-CoV-2 cellular immune response was low and only detectable in one vaccinated KTX recipient. This finding is in line with a recent Belgian study reporting a detectable anti-SARS-CoV-2 T-cell response in approximately 30% of KTX recipients upon BNT162b2 vaccination [38]. Consequently, our data show that both cellular and humoral immune response to SARS-CoV-2 vaccination appear to be diminished in KTX recipients, highlighting the need for additional measures to protect this vulnerable group. The first data following a third dose of mRNA vaccines in KTX recipients showed a serological response in about half of the non-responders to the two-dose schedule [39]. Besides booster immunization, our data support considerations that pausing MMF/MPA prior to anti-SARS-CoV-2 vaccination may further benefit a protective immune response [36,39]. Notwithstanding, KTX recipients should maintain general hygiene measures and alternative vaccination strategies should be considered [34].

Taken together, our results indicate that dialysis patients have a seroconversion rate similar to healthy control groups. A double vaccination with mRNA-1273 generated higher SARS-COV2 IgG titers in this population than the other examined vaccines. Whether this finding is associated with a lower infection rate or a milder course in the case of COVID-19 needs to be further investigated.

The seroconversion in KTX patients was significantly impaired compared to healthy control groups, and cellular or humoral immune responses after two vaccinations vanished rapidly within 2–3 months. Therapy with MMF/MPA proved to be an independent risk factor for missing seroconversion. 

Overall, our data underline the urgent need for a third vaccination and provide first indications for the possible positive effect of the modification of immunosuppressive therapy.

## Figures and Tables

**Figure 1 microorganisms-10-00004-f001:**
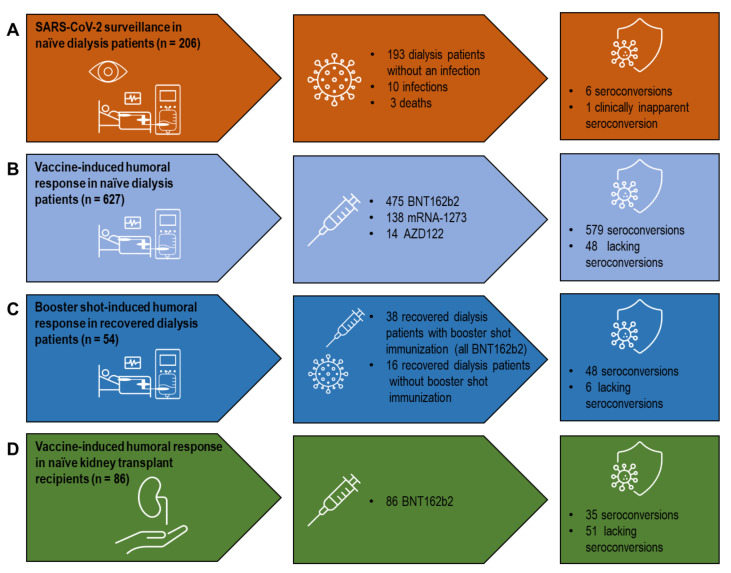
A systematic analysis of vaccine- and infection-induced humoral SARS-CoV-2 immune responses in dialysis patients and kidney transplant recipients. (**A**) SARS-CoV-2 surveillance in SARS-CoV-2-naïve dialysis patients. (**B**) Vaccine-induced humoral response in SARS-CoV-2-naïve dialysis patients. (**C**) Booster shot-induced humoral response in SARS-CoV-2 convalescent dialysis patients. (**D**) Vaccine-induced humoral response in SARS-CoV-2-naïve kidney transplant recipients. Vaccines used: BNT162b2 (BioNTech/Pfizer, Mainz, Germany/New York, NY, USA), mRNA-1273 (Moderna, CA, USA), AZD1222 (AstraZeneca/University of Oxford, Cambridge/Oxford, UK). SARS-CoV-2: severe acute respiratory syndrome coronavirus type 2.

**Figure 2 microorganisms-10-00004-f002:**
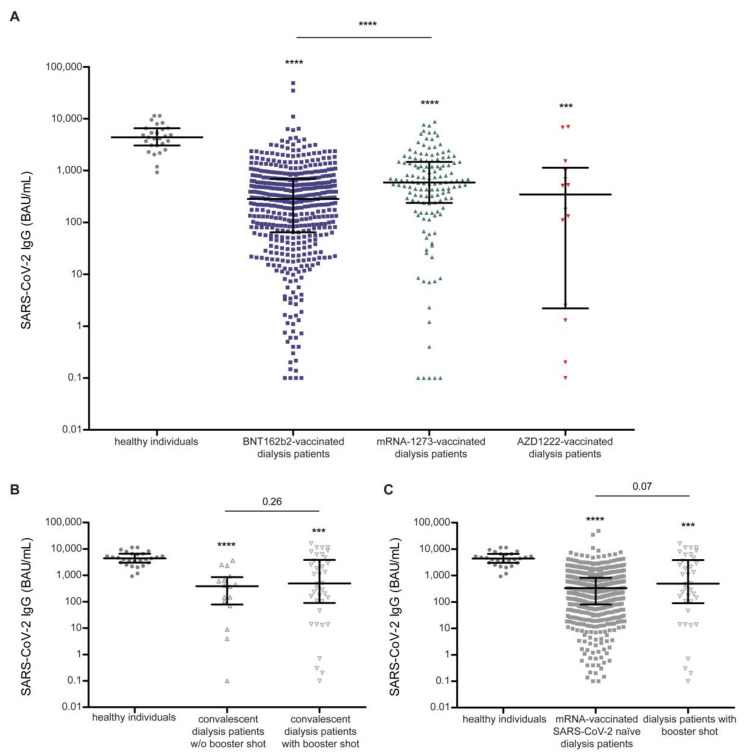
Vaccine- and infection-induced humoral response quantified by anti-SARS-CoV-2 IgG titers in dialysis patients compared to vaccinated healthy individuals. (**A**) Vaccine-induced anti-SARS-CoV-2 IgG titers in dialysis patients differ significantly between dialysis patients immunized by BNT162b2, mRNA-1273, and AZD1222 compared to the reference cohort of mRNA-1273 vaccinated healthy individuals. (**B**) Infection induced anti-SARS-CoV-2 IgG titers in convalescent dialysis patients with (w) and without (w/o) booster shot immunization using BNT162b2 mRNA are reduced compared to vaccine-induced titer in healthy individuals. (**C**) Booster shot-induced anti-SARS-CoV-2 IgG titers in SARS-CoV-2-recovered dialysis patients are elevated compared to mRNA vaccine-induced titers in SARS-CoV-2-naïve dialysis patients. Vaccines used: BNT162b2 (BioNTech/Pfizer, Mainz, Germany/New York, NY, USA), mRNA-1273 (Moderna, CA, Unites States), and AZD1222 (AstraZeneca/University of Oxford, Cambridge/Oxford, UK) in dialysis patients; mRNA-1273 (Moderna, CA, Unites States) in healthy individuals. A Kruskal-Wallis test across all groups was used to indicated differences to healthy individuals. Mann–Whitney U test was used to indicated differences between two groups (*** *p* < 0.001 and **** *p* < 0.0001). Data are represented as median ± interquartile range and each patient is depicted by a single dot. IgG: immunoglobulin G; BAU: binding antibody unit; SARS-CoV-2: severe acute respiratory syndrome coronavirus type 2.

**Figure 3 microorganisms-10-00004-f003:**
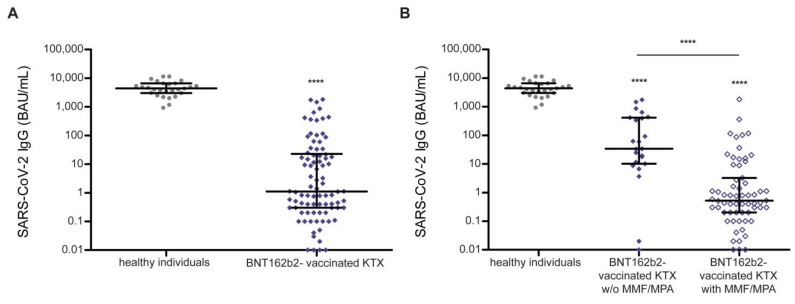
Vaccine-induced humoral response quantified by anti-SARS-CoV-2 IgG titers in kidney transplant recipients and healthy individuals. (**A**) Vaccine-induced anti-SARS-CoV-2 IgG titers in kidney transplant recipients are impaired compared to healthy individuals. (**B**) Vaccine-induced anti-SARS-CoV-2 IgG titers are significantly elevated in kidney transplant recipients with immunosuppression lacking mycophenolate mofetil/mycophenolic acid. Vaccines used: BNT162b2 (BioNTech/Pfizer Mainz, Germany/New York, NY, USA) in kidney transplant recipients, and mRNA-1273 (Moderna, CA, United States) in healthy individuals. A Kruskal–Wallis test across all groups was used to indicate differences with respect to healthy individuals. A Mann–Whitney U test was used to indicate differences between two groups (**** *p* < 0.0001). Data are represented as median ± interquartile range and each patient is depicted by a single dot. IgG: immunoglobulin G; BAU: binding antibody unit; KTX: kidney transplant recipient; MMF: mycophenolate mofetil; MPA: mycophenolic acid; SARS-CoV-2: severe acute respiratory syndrome coronavirus type 2.

**Table 1 microorganisms-10-00004-t001:** Seroconversion and anti-SARS-CoV-2 IgG titer as determined by anti-SARS-CoV-2 serology in dialysis patients vaccinated with BNT162b2 (BioNTech/Pfizer), mRNA-1273 (Moderna), or AZD1222 (AstraZeneca/University of Oxford) and in COVID-19-recovered dialysis patients with/without BNT162b2 booster shot immunization. COVID-19: coronavirus disease 2019; SARS-CoV-2: severe acute respiratory syndrome coronavirus type 2.

Vaccine Used in SARS-CoV-2- Naïve Dialysis Patients	Seroconversion, *n* (%)	Anti-SARS-CoV-2 IgG (BAU/mL), Median (IQR)
BNT162b2 (BioNTech/Pfizer), *n* = 475	440 (92.6)	283.5 (628.5)
mRNA-1273 (Moderna), *n* = 138	129 (93.4)	596.6 (1234.2)
AZD1222 (AstraZeneca/University of Oxford), *n* = 14	10 (71.4)	345.1 (1128.2)
**COVID-19 Recovered Dialysis Patients**		
No booster shot immunization, *n* = 16	14 (87.5)	370.3 (565.5)
Booster shot (all BNT162b2, BioNTech/Pfizer), *n* = 38	34 (89.5)	1103.0 (4570.0)

**Table 2 microorganisms-10-00004-t002:** Baseline characteristics of SARS-CoV-2-naïve dialysis patients with determined anti-SARS-CoV-2 serology after vaccination or infection. ADPKD: autosomal dominant polycystic kidney disease; CAPD: continuous ambulatory peritoneal dialysis; IQR: interquartile range; SARS-CoV-2: severe acute respiratory syndrome coronavirus type 2; SD: standard deviation; yrs: years.

Baseline Characteristics	Seroconversion *n* = 579	No Seroconversion *n* = 48	*p*-Value
Age, yrs (± SD)	68 (± 15)	69 (± 16)	0.823
Sex, *n* (%)			
Female	218 (37.7)	23 (47.9)	0.160
Pre-existing kidney disease, *n* (%)			
Glomerulonephritis	109 (18.8)	8 (16.7)	0.712
Diabetic nephropathy	101 (17.4)	6 (6.3)	0.045
Hypertensive nephropathy	143 (24.7)	9 (18.8)	0.355
ADPKD	41 (7.1)	3 (6.3)	0.828
Other genetic nephropathy	62 (10.7)	12 (25.0)	0.003
Unknown/other	103 (17.8)	12 (25.0)	0.215
Haemodialysis, *n* (%)	559 (96.5)	47 (97.9)	
Years of Haemodialysis (IQR)	4 (2–6)	3 (1–6)	0.343
CAPD, *n* (%)	20 (3.5)	1 (2.1)	
Years of CAPD (IQR)	2 (1–4)	2	0.219

**Table 3 microorganisms-10-00004-t003:** Baseline characteristics of SARS-CoV-2-naïve kidney transplant recipients with determined anti-SARS-CoV-2 serology after vaccination with BNT162b2 (BioNTech/Pfizer). AB0i: AB0 incompatible; ADPKD: autosomal dominant polycystic kidney disease; IQR: interquartile range; MMF: mycophenolate mofetil; MPA: mycophenolic acid; mTOR: mechanistic target Of rapamycin; SD: standard deviation; yrs: years.

Baseline Characteristics	Seroconversion *n* = 34	No Seroconversion *n* = 52	*p*-Value
Age, yrs (± SD)	60.5 (± 15)	63 (± 13)	0.090
Sex, *n* (%)			
Female	16 (47.1)	23 (44.2)	0.504
Immunosuppression, *n* (%)			
Tacrolimus	26 (76.5)	42 (80.8)	0.632
Cyclosporine A	5 (14.7)	9 (17.3)	0.749
Belatacept	0	1 (1.9)	0.416
mTOR	4 (11.8)	3 (5.8)	0.320
Azathioprine	3 (8.8)	3 (5.8)	0.587
MMF/MPA	15 (44.1)	48 (92.3)	<0.001
Steroid	30 (88.2)	50 (96.2)	0.159
Pre-existing kidney disease, *n* (%)			
Glomerulonephritis	9 (26.5)	19 (36.5)	0.330
Diabetic nephropathy	5 (14.7)	3 (5.8)	0.163
Hypertensive nephropathy	1 (2.9)	1 (1.9)	0.759
ADPKD	5 (14.7)	4 (7.8)	0.299
Other genetic nephropathy	6 (17.6)	14 (26.9)	0.319
Unknown/other	8 (23.5)	10 (19.2)	0.632
Kidney transplantation mode, *n* (%)			
Deceased	16 (47.1)	25 (48.1)	0.926
Living donor	16 (47.1)	24 (46.2)	0.934
Living donor AB0i	2 (5.9)	3 (5.8)	0.983

## Data Availability

Data supporting the reported results will be provided from the corresponding authors upon reasonable request.

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
