# Peer review of "Immune Responses to SARS-CoV-2 Infection and Vaccination in Dialysis Patients and Kidney Transplant Recipients"

_microorganisms, 2021, doi:10.3390/microorganisms10010004_

Round 1
Reviewer 1 Report
It was a pleasure to review the article submitted by Affeldt et al. This manuscript was particularly interesting and covered a topic that still needs a lot of studies to better understand SARS-CoV-2 infection and better manage patients.
The article is pleasant to read, clear and well presented. The introduction sets the context and the purpose of the study, using relevant references.
Basing on a large population of dialysis patients and transplant recipients, and a large collection of serum samples, the authors studied the efficiency of different modes of immunization (natural +/- boost, double dose vaccine) and different vaccines.
They showed that seroconversion in dialysis patients depended on the type of vaccine that have been used. Moreover seroconversion was less frequent than in the general population. In kidney transplant recipients, their data showed an only 39.5% seroconversion rate after 2 doses of BNT162b2 vaccine.
Interestingly, they separated patients with and without seroconversion and compared their immunosuppressive regimens. In this way, they demonstrated a difficulty to seroconvert in case of treatment with mycophenolate-mofetil / mycophenolic acid.
Another strength of the study is the prospective analysis of cellular and humoral immunity in KTX recipients. The authors showed a very low induction of immunity in the group (3/32), even lower than the observation in the retrospective study part.
Another important aspect of the article is the antibody titers to SARS-CoV-2 in the different patient groups. Although the authors' conclusions are supported by the results of the study, the methodology used may include biases that may call these results into question.
Indeed, depending on the study periods, antibody titers have been quantified by 2 different assays (IDK or Abbot) that do not target the same antigen (S1 or RBD) and are therefore not perfectly equivalent in principle. However, the authors pooled the different quantitative results from these two assays within the groups, without underlying the distribution of these two types of results.
They justified this choice by comparing the titers obtained by the 2 assays in a small population of dialysis patients (but not in transplant patients or healthy individuals). The results are indeed correlated, but the linear regression is not graphically represented, which does not enable the validation of the correlation. In addition, the medians differ (line 234) and no statistical test is mentioned. They discussed this methodology by citing a previous study (#31) but do not apply the methodology and statistical analyses of this cited study. In addition, in reference #31, the authors compared different quantitative anti-SARS-COV-2 antibody assays, but not the ones in the present article (IDK vs Abbott). In article #31, differences of up to 12-60% between assays are reported, which may greatly influence the results and therefore the conclusions of this article.
#1 : Some additional analyses (major revisions) are therefore necessary to ensure the veracity of the presented results , in particular by referring to the statistical methods of reference #31:
- Plotting the correlation with linear regression.
- evaluating the % difference between the two assays (Bland Altmann, Passing Bablok)
- possibly performing a paired t-test of the values obtained with the two techniques.
If a previous published studied compared the two assays, a simple citation would be sufficient.
If equivalence between the assays cannot be assessed, the titers analyses should be removed from the manuscript or compared depending on the assays in subgroups.
I also noticed some minor revisions to take in consideration:
- #2 Lines 102-104 : is the inclusion period for the cohort is strictly the same as the retrospective part ? It should be specified to understand the risk of exposure to the infection, which is relative to the dynamics of the pandemic.
- #3 Line 147 : the Supplemental Table 1 does not bring useful information. It should be removed or enriched with the same information than the Supplemental Table 2.
- #4 Figure 1 :
- A : “3 deaths” and “1 clinical inapparent seroconversion” should have an alignment shift to underly the cases in the upper counts.
- C : the group contain 54 patients but the authors mentioned “50 seroconversion” (please add an “-s” by the way) and “6 lacking seroconversion” (plural is missing too) = 56 (?!)
- #5 Table 1 : Are the AZD1222 results really relevant as only 14 patients are included comparing to the other groups ? Moreover, the result is not mentionned in the text (§3.2). I suggest a description of the result in the main text, or removing the data from the manuscript (or only in the discussion part ?).
- #6 Lines 199-201 : once again, the relevance of the results from the AZD1222 group is questionable
- #7 Figure 2B and lines 225-227: healthy individuals are not the right control so the term “reference cohort” is confusing. They should have been “convalescent healthy individuals w/ or w/o booster”. It should be highlighted in the §3.3 and discussed.
- #8 Line 262 : the correlation is barely significant (=0.05 and not <0.05 as mentioned in Methods) and we have no information about the positivity or the negativity of the correlation. Therefore, the result is not interpretable. Linear regression plotting should be included in the manuscript, or at least in Supplemental Data (depending on the Editor’s recommendations)
- #9 §3.6 : the CD8+ cellular response results are not mentioned at all. I guess they were all negative but it should be specified in the text
- #10 line 348 : As there is no control group in the cohort, the immune response cannot be described as “impaired” or “diminished” but only “low”, basing on the present study. However, these results can be compared to other publications to support the conclusion of the authors. Indeed, a <10% seroconversion is obviously lower than general population seroconversion.
Reviewer 2 Report
The authors analyzed the immune responses against SARS-CoV-2 in dialysis patients and kidney transplant recipients. This is a important data to further develop prophylactic methods for patients receiving dialysis. Methodology and conclusion look correct. I would recommend it for acceptance after the minor point below.
Pleas confirm the number of patients.
Line 28-29: 833 dialysis patients and 86 KTX recipients
Lines 75-76: 681 HD patients and 86 KTX recipients
Round 2
Reviewer 1 Report
Affeldt et al. revised their manuscript according to the reviewers'comments. The response was clear and scientifically relevant. The revised manuscript is significantly improved comparing to the first draft. This study will bring a lot of knowledges to the community